# Wef-GNN: A Generalizable Graph Neural Network for Crystalline Material Property Prediction

## Abstract

Graph neural networks (GNNs) have shown great promise for predicting properties of crystalline solids. However, existing models struggle to generalize across crystals of varying sizes, and there is a lack of high-fidelity *ab initio* training data. Here, Weighted energy functional GNN (Wef-GNN) addresses the problem of generalizability by introducing a multi-head temporal attention mechanism in the graph update function and a crystalline graph representation scheme that is more size-agnostic compared to the traditional primitive unit cell-based graph representation. Further, it was found that a single Wef-GNN layer can be recycled for all graph convolution steps without considerable loss in accuracy; this leads to deep receptive fields without additional parameters. Wef-GNN outperforms all prior models in a standard band gap prediction benchmark while having much fewer parameters. To address the challenge of high quality *ab initio* training data, a high-fidelity dataset was curated by performing $10,522$ high-accuracy Density Functional Theory (DFT) calculations. Wef-GNN was pre-trained on a standard large dataset of lower-accuracy DFT calculations then fine-tuned with the high-accuracy DFT dataset. The resulting model matches experimental band-gap values much better than other GNNs, and even outperforms the underlying low-accuracy DFT calculations.

## 1 Introduction

In recent years, there has been an explosive demand of new materials for applications ranging from computer chips and solar cells to batteries and more. Meeting this demand requires the discovery of new materials using high throughput screening. At present, the leading tool for preforming high throughput screening of materials is first principles density-functional theory (DFT) calculation. For example, Yuan et al. (2024) used DFT to screen $40,000$ crystals to isolate a good solar cell candidate. Although effective, this approach faces several problems. For instance, DFT's large computational cost makes calculations slow and expensive, restricting exhaustive searches and leaving the vast majority of chemical space unexplored. Another problem is DFT's intrinsic error; Yuan et al. (2024) used the widely adopted Perdew-Burke-Ernzerhof (PBE) DFT functional. PBE has a band gap error of $\sim 0.6$ eV (Jain et al., 2011). Band gap is a crucial property of for solar cells, where a value of $1.1 - 1.6$ eV is typically required, thus a $\sim 0.6$ eV error risks the inadvertent rejection of many promising materials. There exist more accurate DFT functionals, such as Heyd–Scuseria–Ernzerhof (HSE) hybrid functional which has an error of $\sim 0.37$ eV, but these come at even larger computational costs, making high throughput screening use HSE and other higher accuracy functionals not feasible (Garza & Scuseria, 2016). This paper aims to address the speed and band gap accuracy limitations of DFT, with the goal of making a tool better suited for high throughput discovery of new solar cell.

Over the past decade, machine learning (ML) techniques have been perused to solve DFT's limitations in high throughput screening, and have been used for crystal property prediction, inverse design of drug molecules, and more (Reiser et al., 2022; Coley et al., 2017). Early on, representing molecules as SMILES (simplified molecular-input line-entry system) strings (Weininger, 1988) enabled the adoption of machine translation networks such as sequence-to-sequence models (Liu et al., 2017; Schwaller et al., 2018; Wang et al., 2023) and transformers (Vaswani et al., 2017) for predicting the properties and chemical reactions of organic molecules (Schwaller et al., 2019; Tetko et al.,

2020; Irwin et al., 2022; Kassa et al., 2023). However, there is a limitation in using one-dimensional SMILES strings represent three-dimensional molecules, and this limitation that becomes more evident in the case of crystalline solids, which have added complexities such as periodic boundary conditions (PBCs) and disorders (Schütt et al., 2018; Klicpera et al., 2021; Schütt et al., 2021). Graph Neural Networks (GNNs) such as Graph Convolutional Networks (GCNs) (Kipf & Welling, 2017) and Graph Attention Networks (GATs) (Veličković et al., 2018) overcome this limitation by representing molecules and crystals as graphs, thereby better capturing the three-dimensional structural and relational information. This can be seen in the improved performance of graph based models such as Wang et al. (2023), Rasmussen et al. (2023), and Gilmer et al. (2017) in organic molecule prediction tasks over SMILEs based models. It is important to distinguish this domain from Machine Learning Interatomic Potentials (MLIPs) (e.g., NequIP, MACE). While MLIPs have achieved high fidelity using equivariant architectures, they primarily target potential energy surfaces and forces for molecular dynamics, typically utilizing PBE or SCAN level data. They do not address the specific challenge of predicting electronic band gaps, where the underlying DFT functional error (PBE) remains the primary bottleneck.

In materials science, however, GNNs have seen slower adoption (Reiser et al., 2022). Pioneering works like the Crystal Graph Convolutional Neural Network (CGCNN) (Xie & Grossman, 2018) and the MatErials Graph Network (MEGNet) (Chen et al., 2019) have demonstrated the ability of GNNs in predicting formation energies, band gaps, and other properties of crystalline solids with near *ab initio* accuracy, but at a fraction of the computational cost. For instance, Gilmer et al. (2017) have demonstrated Message Passing Neural Networks (MPNNs), can predict material properties up to $10,000\times$ faster than traditional DFT calculations. However, despite these findings, the application of GNNs in materials science has largely remained an intellectual exercise, with limited real-world implementation. The first problem is the lack of generalizibility in GNNs, meaning GNNs struggle to accurately predict the properties of materials with differing crystal sizes; a GNN trained on small crystals will exhibit low accuracy in predicting the properties of larger crystals (Reiser et al., 2022). The second challenge is low accuracy inherited from the PBE-level DFT dataset. Current ML models in materials science are primarily trained on datasets of PBE-level DFT calculations, such as Open Quantum Materials Dataset (Kirklin et al., 2015; Saal et al., 2013) and Materials Project (Jain et al., 2013). Since PBE itself has a band gap calculation error of $\sim$0.6 eV, models trained on this dataset inherit PBE's inaccuracy. While an HSE dataset would mitigate this problem, given its improved accuracy ($\sim$0.37 eV band gap error), the high computational cost of HSE calculations makes generating a sufficiently large dataset infeasible. Recent transformer-based approaches such as Matformer (Yan et al., 2022) and PotNet (Lin et al., 2023) have advanced the field. However, many rely on pre-training with experimental data to mitigate DFT bias (Song et al., 2024; Das et al., 2023). It is argued here that for high-throughput photovoltaic screening, relying on experimental data is insufficient due to the thermodynamic inconsistency between 0K DFT predictions and finite-temperature experimental measurements, as well as the scarcity of experimental band gap records ($< 4k$) compared to the vast chemical space.

This paper attempts to address the challenges faced by GNNs in material science. First, Weighted energy functional GNN (Wef-GNN) is presented, an architecture that mitigates the generalizibility issue of GNNs by introducing a multi-head temporal attention mechanism, GNN layer recycling, and a new graph representation scheme for crystal structures. Note that Wef-GNN utilizes a hybrid attention strategy: static (spatial) attention is applied during to weigh neighbor importance, while temporal attention is applied to weigh a node's historical states. Next, a hybrid training approach is introduced, where a large dataset of PBE calculations is used for pretraining and a smaller dataset of HSE06 (HSE with $25\%$ mixing parameter) calculations is used for fine tuning. It is shown that a model trained with this approach achieves band gap accuracies that surpass that of DFT computed using the PBE functional.

## 2 WEF-GNN ARCHITECTURE

### 2.1 GRAPH REPRESENTATION OF CRYSTALS

A single unit cell of periodic crystal can be represented as a graph $\mathcal{G} = (V, E)$, where each node $i \in V$ is assigned a feature vector $x_i$. Initially, works such as Xie & Grossman (2018) used several atomic descriptors, like atomic number, atomic mass, electronegativity, for $x_i$. However, Chen et al.

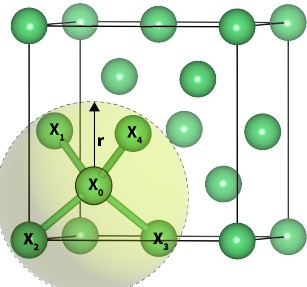

Figure 1: Diamond-cubic silicon with a 0.235 nm nearest-neighbor separation. Atom $x_0$ has edges with all atoms within a cutoff radius of $r_c < 0.30$ nm.

(2019) observed that comparable accuracy can be achieved using atomic number alone. Here, it is found that using a combination of atomic number and fractional coordinates improves performance. Incorporating fractional coordinates helps differentiate identical atoms by giving them distinct, intensive, signatures based on their location in the unit cell, and leads to a more expressive model. Thus, $x_i$ is constructed by embedding the atomic numbers using a learnable matrix and concatenating this with a Gaussian basis expansion of the fractional coordinates (Gaussian basis expansion acts as a smooth analogue of one-hot encoding for continuous inputs).

Next, an edge between two nodes, $i$ and $j$, is represented using the feature vector $e_{ij}$ that stores the interatomic distance between the nodes. Several rules have been proposed for determining if an edge exists between two nodes, such as fixed cutoff radius, $k$-nearest neighbors, and Voronoi cells (Ruff et al., 2024). This work determines the existence of an edge using a cutoff radius, where nodes $i$ and $j$ are connected by the edge $e_{ij}$ if their separation distance is below the cutoff radius $r_c$. Here, the cutoff radius approach is preferred because of the strength of the interaction of a pair of atoms is more dependent on their separation distance than the number of neighbors a given atoms has. This can be easily observed from the interatomic potential of a set of atoms (Scott et al., 1991) or, more simply, the Lennard-Jones potential (Jones & Chapman, 1924), where the force of attraction between atoms has an inverse power relation with the separation distance.

Given the crystal structures are periodic repeats of the unit cell, lattice periodicity needs to be considered when determining a node's neighbors. Therefore, PBCs are imposed by evaluating the minimum image distance. To see why this is necessary, consider a unit cell of silicon (Si), where each atom in the unit cell has a neighbor 0.235 nm away. Choosing $r_c < 0.30$ nm gives exactly four neighbors for each atom. This is true whether the atom resides within the unit cell interior or on one of the corners (Figure 1); thus it is essential to reflect this periodicity in the distance calculations. This is done by calculating separate interatomic distances after shifting along each Cartesian axis by -1, 0, and 1 one at a time. Then the minimum from all these distance calculations is used as the distance ($d$) between a pair of nodes, i.e., for a pair of nodes $i$ and $j$, $d_{ij} = \min_{\tau \in \{-1,0,1\}^3} \| L(f_j - f_i + \tau) \|_2$ ,where L represents the lattice vectors and f the fractional coordinates.

## 2.2 GNN ARCHITECTURE

Similar to other GNNs (Gilmer et al., 2017; Chen et al., 2019), Wef-GNN has three main parts: AGGREGATION, UPDATE, and READOUT. Each layer of a GNN (each message-passing step), made up of the AGGREGATION and UPDATE, receives the node and edge feature vectors $x_i^t$ and $e_{ij}^t$ and produces new feature vectors $x_i^{t+1}$, and $e_{ij}^{t+1}$. During AGGREGATION, the node features of the neighbors of node $i$, i.e., $\mathcal{N}(i)$, get aggregated into a single feature vector $x_i^*$. UPDATE then combines $x_i^*$ with $x_i^t$, yielding $x_i^{t+1}$. After $T$ message passing steps, a permutation-invariant READOUT is used to produce a final graph level embedding, $\mathcal{G}$, for downstream tasks .

While Wef-GNN uses attention during AGGREGATION similar to Veličković et al. (2018), what differentiates Wef-GNN is the introduction of a multi-head temporal attention mechanism during UPDATE. The multi-head temporal attention mechanism lets each node attend to its own history, $\{x_i^K, \ldots, x_i^{t-1}\}$, where $K$ is the look back window (how far back in its history the node looks).

---

**Algorithm 1** Wef-GNN Training

---

**Input**: $x^{t=0}$
**Output**: $\mathcal{G}$
1: **for** $i$ in $\{x^{t=0} \mid x^{t=0} \in nodes\}$ **do**
2:     **for** $j$ in $\{x^{t=0} \mid x^{t=0} \in nodes,\ x^{t=0} \neq i\}$ **do**
3:         $d_{ij} = \min_{\tau \in \{-1,0,1\}^3} \parallel \mathrm{L}(\mathrm{f}_j - \mathrm{f}_i + \tau) \parallel_2$
4:         **if** $d_{ij} < r_c$ **then**
5:             Add $[i, j]$ to edges
6:         **end if**
7:     **end for**
8: **end for**
9: Let $t = 0$.
10: **while** $\{t < T\}$ **do**
11:     $x_i^*, e_{ij}^* = \textsc{Aggregation}[x_i^t, e_{ij}^t]$
12:     $x_i^{t+1} = \textsc{Update}[x_i^*, x_i^t]$
13:     $e_{ij}^{t+1} = \textsc{Update}[e_{ij}^*, e_{ij}^t]$
14:     $t = t + 1$
15: **end while**
16: $G = \textsc{Readout}[x_i^T]$
17: **return** $\mathcal{G}$

---

Let us now discuss the components of $\textsc{Aggregation}$ in Wef-GNN in more detail. As explained previously, at $t = 0$, $x_i$ is the concatenation of a learned embedding of the atomic number and a Gaussian basis expansion of the fractional coordinates (centered between 0 and 1), while $e_{ij}$ is the interatomic distance between $i$ and $j$ expanded to a Gaussian basis (centered between 0 and $r_c$). During $\textsc{Aggregation}$, $x_i^t$ and $e_{ij}^t$ first undergo weighed linear transformations to give $x_i^m$ and $e_{ij}^m$, where $x_i^m = x_i^t W_a + b_a$ and $W_a$ and $b_a$ are learned matrices; likewise, $e_{ij}^m = e_{ij}^t W_e + b_e$. Now, let us define:

$$s_{ij} = e_{ij}^m \parallel (x_i^m + x_j^m)$$

where $\parallel$ represents concatenation. Linearly transforming $s_{ij}$ yields $s_{ij}^* = s_{ij} W_s + b_s$. Next, the spatial attention is added as follows:

$$s_{ij}^m = s_{ij} \cdot \alpha_{ij} \cdot \alpha_{ji}$$

where $\alpha_{ij}$ and $\alpha_{ji}$ are attention coefficients given by:

$$\alpha_{ij} = \frac{\exp(A_e \cdot (x_i^m \parallel x_j^m \parallel e_{ij}^m))}{\sum\limits_{k=1}^{q} \exp(A_e \cdot (x_i^m \parallel x_k^m \parallel e_{ik}^m))}$$

where $A_e$ is a learned matrix, and $q$ is the number of neighbors of node $i$. Because the graphs are represented using the coordinate format (COO), multiplying $s_{ij}^m$ by $\alpha_{ij}$ and $\alpha_{ji}$ ensures order invariance. Therefore, for each neighbor $j$ of $i$, $s_{ij}^m$ contains information about $e_{ij}$, $j$, and the relevance of $j$ to $i$. Thus, the information about all the neighbors of node $i$ can be gathered as follows:

$$h_i = \sum_{k=1}^{q} s_{ik}^m + s_{ki}^m$$

$$x_i^* = \mathrm{PReLU}[(x_i^m \parallel h_i)W_s]$$

where PReLU is a parametric rectified linear unit, serving as a nonlinear activation function. And similarly, $e_{ij}^* = \mathrm{PReLU}[s_{ij}^m]$.

Finally, $\textsc{Update}$ combines $x_i^t$ and $e_{ij}^t$ with the intermediate values $x_i^*$ and $e_{ij}^*$, yielding $x_i^{t+1}$ and $e_{ij}^{t+1}$. Many $\textsc{Update}$ functions have been proposed in literature, such as Gated Recurrent Unit (Cho et al., 2014), summations and other non-weighted techniques (Gilmer et al., 2017). In Wef-GNN, a multi-head temporal attention is introduced and leads to better performance on crystalline material

benchmarks. Bahdanau (Bahdanau et al., 2015) style cross attention ($\gamma_{temp}$) was used to combine the $x_i^*$ and $e_{ij}^*$ with the history of the node and edge states:

$$x_i^{t+1} = \gamma_{\text{temp}}\big(\text{query} = x_i^*, \text{ key} = x_i^*, \text{ value} = \{x_i^{t-1}, \ldots, x_i^{t-K}\}\big)$$

where $K$ is the look back window. A similar temporal attention is applied to obtain $e_{ij}^{t+1}$. The rest of the multi head attention mechanism follows (Vaswani et al., 2017) and uses concatenation for head aggregation. Layer normalizations, residual connections, and dropouts are used, but omitted have been here for brevity.

After $T$ message passing steps, the final node features are pooled by READOUT to obtain the graph-level representation $\mathcal{G}$. Although various READOUT functions such as Set2Set (Vinyals et al., 2016), transformer encoders and others (Gilmer et al., 2017) have been proposed, here, simply applying a non-linearity followed by component-wise mean has been found to match their accuracy while being much less computationally demanding. The full Wef-GNN framework is summarized in Algorithm 1.

A weight recycling strategy is employed where the same learnable matrices ($W_a, W_e, A_e$) are reused at every message passing step $t$. This design choice is motivated by physical intuition: the fundamental laws governing interatomic interactions (e.g., electrostatic forces, covalent bonding) are invariant to the "depth" of the iteration. By sharing weights across layers, the GNN is forced to learn a universal interaction function that iteratively refines the crystal representation towards a stable equilibrium, conceptually similar to the self-consistent field (SCF) iterations used in DFT calculations. This contrasts with standard deep GNNs where distinct layers act as hierarchical feature extractors. Furthermore, this recurrence allows Wef-GNN to stack more layers (deepening the receptive field to capture long-range interactions) without the risk of parameter explosion or overfitting, as the parameter count remains constant regardless of depth. Regarding invariance, Wef-GNN is invariant to rotation and translation by design, as edge features $e_{ij}$ are derived solely from interatomic distances, and node features $x_i$ are derived from atomic numbers and Gaussian-expanded fractional coordinates which capture the local environment relative to the unit cell basis.

## 3 ADDRESSING GENERALIZABILITY

One of the major challenges GNNs face has to do with generalizability. Stacking many GNN layers—performing too many message passing steps—causes node representations to become indistinguishable, resulting in a performance decline known as over-smoothing. The number of message passing steps that result in such over-smoothing depends on the size of graph. Given that different materials can have very different unit cell sizes, this problem has hindered a single GNN from generalizing to different material systems.

For example, consider the materials Si and AgIrF$_7$. The primitive unit cell of Si has two atoms, each having a degree of 1 when using $r_c < 0.3$ nm; after only one message passing step, each node has already incorporated information from all the nodes in the graph. Subsequent message passing steps merely homogenize the features, reducing model performance. On the other hand, the primitive unit cell of AgIrF$_7$ contains 36 atoms, and using the same cutoff $r_c < 0.3$ nm, the average node degree of the crystalline graph is 7.4. As a result, more than one message passing step would be required for the nodes in AgIrF$_7$ to properly incorporate information of their neighborhood. This disparity illustrates the generalizability challenge in applying GNNs across crystals of widely varying size. Larger crystals require many more message-passing steps for high accuracy predictions, while smaller graphs would suffer from this.

It is proposed here that using conventional unit cell representations in place of the commonly used primitive unit cell representations would partially mitigate the size variance problem. The underlying physical infinite lattice is invariant to the choice of unit cell. However, the computational graph topology $\mathcal{G}$ processed by the GNN is not invariant. Specifically, the READOUT operation (global pooling) aggregates information from $|V|$ nodes. A primitive cell graph (e.g., $|V| = 2$) suffers from rapid over-smoothing during the READOUT phase compared to a conventional cell graph (e.g., $|V| = 8$), as the receptive field covers the entire small graph almost instantly. Standardizing on conventional cells ensures a minimum node count (increasing the median from 16 to 21 atoms), which stabilizes the node distribution for the message passing mechanism.

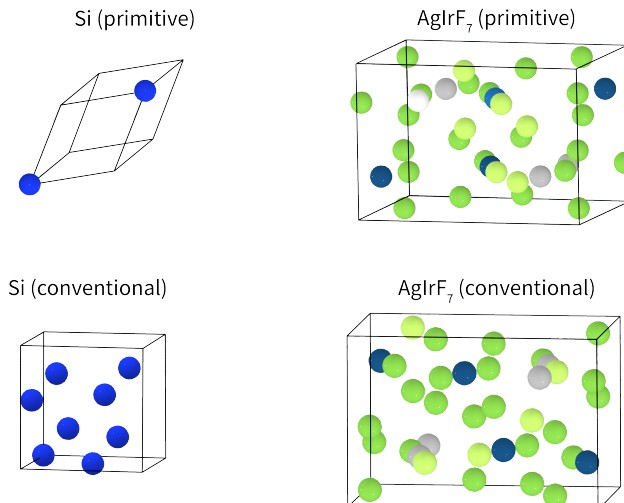

Figure 2: Changing the graph representation of Si from a primitive unit cell made up of 2 atoms to a conventional unit cell with 8 atoms (left). Going from a primitive $AgIrF_7$ unit cell with 36 atoms to a conventional unit cell with 40 atoms (right).

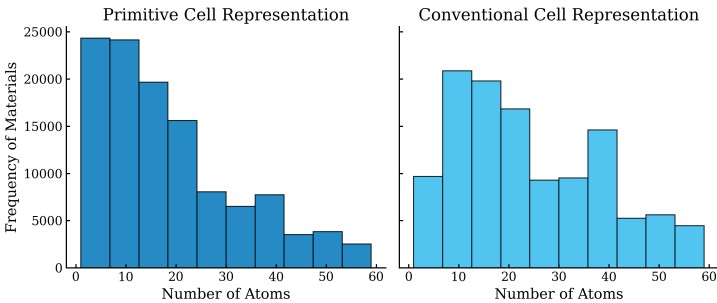

Figure 3: Histogram of the number of atoms for the $\approx$106,000 crystals in the Matbench `matbench_mp_gap` dataset when graphs are built using primitive cell (left) and conventional cell (right) representations. Using conventional cells decreases the interpercentile range while increasing the median number of atoms from 16 to 21, reducing the prevalence of very small graphs that are prone to over-smoothing.

Primitive unit cell representations have often been preferred as they keep graph sizes small. However, this comes at the cost of increasing the size disparity between the smallest and largest crystals. In the Si and $AgIrF_7$ example, switching to conventional unit cell representations increases the size of Si from 2 atoms to 8 atoms, a 4x increase, while it only increases the size of $AgIrF_7$ from 36 atoms to 40 atoms, a 1.1x increase (Figure 2). Accordingly, more message passing steps would lead to less over smoothing in Si, permitting a shared message passing depth. The trade-off is a modest increase in computational cost due to larger graphs, but the benefit is markedly improved generalizability. Consider Matbench's (Dunn et al., 2020) `matbench_mp_gap` dataset which consists $\approx$106,000 crystals from Materials Project (Jain et al., 2013). Switching the crystal representation in `matbench_mp_gap` from primitive cell to conventional cell increases the median number of atoms from 16 (primitive) to 21 (conventional), while decreasing the interpercentile range (P1-P99) from 54 (primitive) to 53 (conventional). This implies that the number of graphs having too few nodes and get over-smoothed is lower, while larger graphs are still being adequately processed (Figure 3).

Finally, it is important to clarify the specific use of the term "generalizability" in this context. While in broader machine learning this term typically refers to performance on out-of-distribution data regarding class labels or chemical space, as explained in Reiser et al. (2022), within the domain

Table 1: Matbench leaderboard for `matbench_mp_gap`. Only the top 5 models are shown here.

| Algorithm | MAE (meV) | Std (meV) |
|---|---|---|
| Wef-GNN | 117.1 | 2.5 |
| Wef-GNN (Recycled) | 117.3 | 2.3 |
| coGN | 155.9 | 1.7 |
| DeeperGATGNN | 169.4 | 4.7 |
| coGNN | 169.7 | 3.5 |
| ALIGNN | 186.1 | 3.0 |
| MegNet (kgcnn v2.1.0) | 193.4 | 8.7 |
| Dummy | 1327 | 6.0 |

of Graph Neural Networks for crystallography, it is used to denote the model's robustness to topological distribution shifts. A dataset dominated by small primitive cells presents a vastly different topological distribution (e.g., mean degree, diameter, node count) than a test case involving large supercells or complex polymorphs, even if the chemical composition is identical. A model that cannot maintain accuracy when the graph size scales—despite the physical material remaining invariant—has failed to generalize its learned interaction rules to larger topologies. By shifting the training distribution toward larger, standardized Conventional Cells, Wef-GNN aligns the training topological distribution more closely with the inference distribution required for complex material simulations, thereby improving this structural generalizability.

## 4 PERFORMANCE

The initial goal of Wef-GNN is to provide a new tool for high throughput screening of solar materials. Accordingly, this section evaluates the performance of Wef-GNN against existing ML models on band gap prediction tasks and the next section provides an extension to the training of Wef-GNN to help it surpass the accuracy of the widely used PBE level DFT calculations. Wef-GNN, was benchmarked on Matbench's `matbench_mp_gap` dataset, which is commonly used to assess ML models for materials science. textttmatbench_mp_gap consists of $\approx$106,000 PBE level band-gap calculations. A Wef-GNN model was built using hidden layer dimensions of 128, 3 message passing steps, 2 attention heads in the multi head temporal attention, an atomic number embedding dimension of 16, and 32, and 512 Gaussian kernels for the fractional coordinate and interatomic distance Gaussian basis expansion. After seven hours of training on Google V4-8 Tensor Processing Unit (TPU) running TensorFlow 2.14.0, the model had a mean absolute error (MAE) of 0.117 eV using the Matbench cross-validation split (MAE is the most informative metric for assessing a model's ability to predict energy based crystal properties). This result places Wef-GNN at the top of the Matbench band gap prediction leaderboard. Notably, the technique of reusing weights for different message passing steps enables Wef-GNN (recycled) to achieve such low errors with only $\approx$420,000 parameters, compared to the $> 1$ million parameters of the next 5 models on the Matbench band gap prediction leaderboard (Table 1).

Let us now assess how well Wef-GNN generalizes to crystals of different sizes. A Wef-GNN model was trained on dataset of mostly small crystals, ones with $\leq 16$ atoms per unit cell. This model achieved a MAE of 0.098 eV on a test set of small crystals. Then model was then evaluated on a test set of crystals having between 32 and 64 atoms per unit cell, where it had a MAE of 0.110 eV, only a 12 meV increase. This shows Wef-GNN has good generalizability for crystals of varying sizes.

### 4.1 ABLATION STUDIES

To validate the architectural choices, two ablation studies analyzing the update mechanism and the graph representation strategy were preformed.

**Impact of Temporal Attention.** The necessity of the Multi-Head Temporal Attention mechanism in the UPDATE step was investigated by replacing it with two standard baselines: a Gated Recurrent Unit (GRU), which is common in Recurrent GNNs, and a simple Multi-Layer Perceptron (MLP) update. The Temporal Attention model achieved an MAE of $117.1$ meV, substantially outperforming the GRU baseline ($141.2$ meV) and the MLP baseline ($148.5$ meV). The superior performance of Temporal Attention over GRU can be attributed to the "vanishing history" problem in standard RNNs. In a crystal graph, the initial node state $x^{t=0}$ contains critical chemical identity information (atomic number), while intermediate states $x^{t=k}$ contain diffused structural information. A GRU tends to overwrite the initial chemical identity with structural noise after several iterations. In contrast, Temporal Attention maintains a direct path to the entire history $\{x_i^{t=0} \ldots x_i^{t-1}\}$, allowing the model to dynamically weigh the importance of the original chemical species versus the aggregated neighborhood environment at every step.

**Impact of Conventional Unit Cells.** The hypothesis that Conventional Unit Cells provide a more robust training topology than Primitive Unit Cells was evaluated. When trained on Primitive Cells, the model's performance degraded significantly, with MAE rising from $117.1$ meV to $154.0$ meV, a $31\%$ increase in error. This validates our hypothesis regarding graph size variance. Primitive cells for simple materials (e.g., metals, simple semiconductors) often contain only 1 or 2 atoms. In such small graphs, the READOUT operation causes immediate over-smoothing, as the receptive field covers the entire graph after a single message-passing step. This prevents the model from learning complex spatial hierarchies. By standardizing on Conventional Cells, a "minimum graph size" is effectively enforced (increasing the median atom count from 16 to 21), ensuring that the message-passing dynamics are consistent across the dataset. This topological consistency is crucial for the weight-shared layers to generalize effectively across diverse crystal systems.

## 5 HYBRID TRAINING

The performance of Wef-GNN on the Matbench band gap prediction task is impressive; however, Matbench's underlying dataset from Materials Project is computed using the PBE level DFT calculations. Given PBE itself has a band gap error of $\sim 0.6$ eV compared to experimental values, no supervised model trained solely on a PBE dataset can outperform this inherent ceiling (Jain et al., 2011). The low band gap accuracy of PBE level DFT calculations is the second limiting factor in high throughput discovery of new solar materials, and also presents the second hurdle GNNs are facing for crystallite material property prediction: lack of a high accuracy training dataset.

As discussed in Section 1, the Hybrid functional HSE06 offers greater accuracy over PBE, with a band gap error of $\sim 0.37$ eV (Komsa & Pasquarello, 2011). While a dataset of HSE06 calculations would help models better understand the underlying physical phenomenon and make better band gap predictions, HSE06 calculations are much more computationally demanding. For instance, performing an HSE06 calculation of an 8 atom unit cell requires approximately 4 hours on an NVIDIA A100 GPU. Therefore, recalculating the $154,000$ materials in the Materials Project using HSE06 is not feasible. To mitigate this problem, it is proposed here that recalculating a small subset of the PBE calculations using HSE06, and using these to fine tune a model pretrained with the $154,000$ PBE calculations would enable GNNs to learn the relationship between the two functionals and utilize the high accuracy of HSE06 and the size of the PBE dataset for more accurate band gap predictions across a wide chemical space.

Accordingly, here, a new dataset was curated where $10,522$ materials from the PBE dataset were recalculated using HSE06 (DFT calculation details are presented in the Supporting Information). Next, a control model, *Model A*, was pre-trained on the full ($154,000$) PBE band gap data from the Materials Project, and it achieved a testing MAE of $0.120$ eV for a PBE ground truth. However, when tested on a dataset of $>4,000$ experimentally measured band gaps curated by Zhuo et al. (2018), *Model A*'s MAE was $0.66$ eV. This result is in line with PBE's known deviation from experimental values of $\approx 0.6$ eV, confirming that the model simply inherits the PBE's limitations.

*Model A* was then fine-tuned on the newly curated HSE06 dataset, yielding *Model B*. When tested on the dataset of $>4,000$ experimentally measured band gaps, *Model B* had a MAE of $0.40$ eV, a $39\%$ improvement over *Model A*, and even more impressively, a $33\%$ improvement over standard PBE-level DFT calculations. Crucially, when validated against a held-out test set of ground-truth HSE06 calculations (not seen during training), *Model B* achieved an MAE of $0.11$ eV. This confirms that the

model has successfully learned the high-fidelity functional landscape and is not merely memorizing the fine-tuning set. Given its higher band gap prediction accuracy compared to PBE level DFT along with its much faster prediction speed, *Model B* is well suited for use in high throughput screening of optimal band gap materials. The band gap accuracy gain in *Model B* was achieved with only 7% of the training data recomputed at using HSE06, demonstrating that full recalculation of the entire dataset using HSE06 is unnecessary. This suggests that a model pre-trained using a dataset of lower accuracy functionals can be fine-tuned using a high accuracy functional dataset, and achieve accuracies close to the high accuracy calculations. This is a step forward in addressing the high accuracy data scarcity problem that has long plagued GNNs and other ML models in materials science. Future work can focus on using DFT functional even more accurate than HSE06 to curate a small finetuning dataset.

## 6 FUTURE PROSPECTS

The success of the hybrid training approach and the Wef-GNN architecture opens several avenues for future research, particularly in the domain of Machine Learning Interatomic Potentials (MLIPs) and geometry optimization. Currently, *ab initio* geometry optimization using high-accuracy functionals like HSE06 is often computationally prohibitive, as it requires calculating forces and stresses at every relaxation step, a cost significantly higher than a single static calculation. Consequently, most material databases rely on PBE-relaxed structures, which may yield inaccurate bond lengths and lattice parameters for strongly correlated systems.

The Wef-GNN framework offers a potential solution to this bottleneck. The weight-recycling mechanism effectively captures long-range interactions essential for accurate force prediction, while the hybrid training strategy—pre-training on abundant PBE structural data and fine-tuning on sparse HSE06 gradients could enable the development of MLIPs that emulate HSE06 accuracy at inference speeds. This would allow for the rapid relaxation of crystal structures into geometries that are more consistent with experimental observations than those obtained via standard DFT. Furthermore, the model's ability to learn high-fidelity electronic properties suggests it could be extended to predict full electronic band structures ($E$-$k$ relationships) and density of states (DOS), moving beyond scalar band gap predictions to provide a comprehensive picture of charge carrier dynamics for photovoltaic applications.

## 7 CONCLUSION

This work introduced Wef-GNN, a size-agnostic Graph Neural Network designed to overcome the generalizability and accuracy limitations prevalent in material property prediction. By shifting from primitive to conventional unit cell representations, the architecture enforces topological consistency across the dataset, mitigating the over-smoothing issues that plague standard GNNs when processing small crystal graphs. The introduction of a Multi-Head Temporal Attention mechanism in the UP-DATE function allows the model to retain critical chemical history against structural noise, while the novel weight-recycling strategy acts as a physical regularizer, enabling deep receptive fields with a minimal parameter footprint ($420\,\mathrm{k}$ parameters). Benchmarking on the `matbench_mp_gap` dataset demonstrated that Wef-GNN achieves state-of-the-art performance with an MAE of $0.117\,\mathrm{eV}$. Furthermore, to transcend the accuracy ceiling of standard DFT, a hybrid training protocol was established using a newly curated dataset of $> 10,000$ HSE06 calculations. This approach successfully bridged the gap between computational availability and physical accuracy, reducing the prediction error against experimental measurements from $0.66\,\mathrm{eV}$ (PBE baseline) to $0.40\,\mathrm{eV}$ (HSE06 fine-tuned). These results validate that integrating physically motivated architectural constraints with multi-fidelity learning strategies is a robust path toward high-throughput, high-accuracy materials discovery.

## 8 REPRODUCIBILITY STATEMENT

The TensorFlow source code of Wef-GNN is made available as part of the supplementary materials. The newly curated HSE06 dataset is made available as part of the supplementary materials, along with details of the DFT Vienna Ab initio Simulation Package settings used for the calculations.

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
