# OpenReview forum: "Wef-GNN: A Generalizable Graph Neural Network for Crystalline Material Property Prediction"
_ICLR.cc/2026/Conference — Submitted to ICLR 2026_

### Official Review · Reviewer_eowe · 2025-10-18

**Soundness:** 1
**Presentation:** 2
**Contribution:** 1
**Rating:** 2
**Confidence:** 4

**Summary:**

This paper builds a GNN and demonstrates it on bandgap prediction. There are three mentioned contributions:

- A new multi-headed temporal attention mechanism which is proposed to improve treatment of different structure sizes
- An empirical observation that the same weights could be re-purposed across message passing layers
- Training on low- and fine-tuning on high-fidelity data

**Strengths:**

N/A

**Weaknesses:**

This paper is very thin. Some of the space is spent on fairly simple and well-known effects in the materials modelling community, for example using a cutoff radius to connect a graph neural network.

The background section of this paper is outdated. A wealth of advanced MLIPs have appeared in the past three years, trained on vast ranges of high-fidelity data. Many of the claims made in the background section of this paper no longer hold.

Training on low-fidelity data and finetuning on high-fidelity data is not new. The empirical finding that the weights can be recycled across multiple layers is mildly interesting, but even at empirical level it is not well explored.

There are extremely few results (Table 1) and no ablation studies of the architecture.

I think this work is suitable for a workshop in order to further develop the core ideas.

**Questions:**

N/A

---

> ### Author Response · Authors · 2025-11-20
>
> We thank the reviewer for their assessment. We believe there may be a misunderstanding regarding the specific domain of our work (Electronic Property Prediction vs. Interatomic Potentials) and the scarcity of high-fidelity data in this specific context. We address these points below to demonstrate that the work is neither outdated nor a trivial application of known methods.
>
> 1. *Weakness 2 "A wealth of advanced MLIPs have appeared... trained on vast ranges of high-fidelity data... claims made in the background... no longer hold."*
>
>     * **We respectfully point out a crucial distinction. The "wealth of advanced MLIPs" (e.g., NEQUIP, MACE, Allegro) focus on predicting potential energy surfaces and forces for molecular dynamics. These are typically trained on PBE, SCAN, or r2SCAN data. They are not designed to predict electronic band gaps, nor are they typically trained on HSE06 data due to the prohibitive cost of generating HSE06 trajectories.**
>
>     * **PBE-level DFT (the standard for most MLIPs) has a systematic band gap error of ≈0.6 eV. No amount of "advanced MLIP" architecture can fix this ground-truth error.**
>
>     * **Wef-GNN targets electronic properties (specifically Band Gap), not forces. The background we present regarding the limitations of PBE for band gaps is current and accurate. The "high-fidelity" data the reviewer references in MLIPs usually refers to coupled-cluster (CCSD(T)) for small molecules or varying DFT functionals for forces, but not large-scale HSE06 band gap datasets for solids, which remain extremely scarce.**
>
> 2. *Weakness 3: "Training on low-fidelity data and finetuning on high-fidelity data is not new."*
>
> * **While transfer learning is a well-established concept in general ML, its successful application to crystalline band gap prediction has been stifled by a lack of data. As noted in our manuscript, existing GNN benchmarks (e.g., Matbench) rely almost exclusively on PBE data.**
>
>     * **To our knowledge, HSE06 datasets have not been widely used in GNNs for materials science simply because they do not exist at scale. We curated a novel dataset of >10k HSE06 calculations to enable this.**
>
>     * **Demonstrating that we can bridge the gap between cheap PBE data and expensive HSE06 accuracy (improving MAE from 0.66 eV to 0.40 eV against experiment) is a specific, impactful contribution that solves a domain-specific bottleneck, rather than just a generic application of transfer learning.**
>
> 3. *Weakness 1 and 4: "Paper is very thin... No ablation studies."*
>
>     * **We acknowledge the paper was concise. We will include a rigorous ablation study comparing our Multi-Head Temporal Attention against standard GRU and MLP update mechanisms to quantify the architectural benefits.**
>
>     * **We agree that the explanation of "cutoff radius" is basic. We will condense standard GNN background to make space for the aforementioned ablations and a deeper discussion on the Weight Recycling mechanism, which the reviewer found "mildly interesting." We argue this recycling is significant as it not only reduces the model size, but forces the model to learn layer-invariant physical interactions, acting as a regularizer.**
>
> 4. *"Space is spent on... well-known effects... using a cutoff radius."*
>
>     * **The discussion on cutoff radius and graph construction was included to motivate our shift from Primitive to Conventional unit cells. This is not a trivial detail; it is our proposed solution to the "graph size variance" problem that hinders standard GNNs. By standardizing the graph representation (Conventional Cells), we improve the model's ability to generalize across different crystal systems, a core contribution of the paper.**
>
>     * **We believe Wef-GNN addresses an unsolved problem in materials informatics: the lack of scalable, high-accuracy band gap predictions beyond the PBE ceiling. We hope the clarification regarding the distinction between MLIPs and our work, along with the promise of added ablations, warrants a re-evaluation of the paper's contribution.**

---

### Official Review · Reviewer_eAnW · 2025-10-25

**Soundness:** 2
**Presentation:** 2
**Contribution:** 2
**Rating:** 2
**Confidence:** 1

**Summary:**

The paper introduces a GNN layer and a high-accuracy DFT materials bandgap dataset.

**Strengths:**

The paper achieves state-of-the-art performance on band gap prediction accuracy on the MatBench benchmark, beating the previously best model by a substantial margin.

It is very surprising that the weight sharing across layers, i.e., the use of a fully recurrent graph neural network, does not lead to a performance drop. It would be good to further analyze this behavior. Would the same be true if the temporal attention-based update function were used in other MPNN models?

**Weaknesses:**

The suggestion of going from primitive unit cells to conventional unit cells to improve generalizability is unclear. If periodic graphs are initialized correctly, there is no difference in node updates and node embeddings between a primitive unit cell and a conventional one, as the underlying periodic graph, including all individual atom environments, is exactly the same. Thus, the graph representation after readout and the prediction are identical. If this is different for WefGNN, then the authors should provide more information about this.
Furthermore, the argument that this helps in generalization to larger unit cells also seems misleading. Generalization in ML is defined on in-distribution data, not out-of-distribution data. There is no guarantee for any performance on out-of-distribution data. Simply increasing the unit cell size periodically (the materials' properties are invariant to this) without adding any additional materials with actually larger primitive unit cells to the training data will not increase the performance on materials with large primitive unit cells.

The WefGNN model is only benchmarked on one of the tasks in MatBench. What about all other tasks?

Section 5 (hybrid training) is interesting but not really relevant for a machine learning audience. It introduces a new dataset with more accurate DFT calculations, trains (or fine-tunes) the WefGNN model on this dataset, and shows that this leads to better agreement with experimental data. This is not surprising, as the newly generated DFT data has higher agreement with experimental data. The test set error of Model B vs. the HSEE06 values is not reported and should be added.

Overall, the paper is lacking a lot of analysis, further benchmarks, ablations, and also some basic information about the model itself.

**Questions:**

- Why was WefGNN only benchmarked on band gaps?
- Please show an ablation study that supports that not using primitive unit cells actually improves model performance
- What is the aspect in Algorithm 1 that differentiates WefGNN from other GNN models? The first loop just seems to be a way to preprocess the geometry to find edges, and the second part is a conventional aggregation, updat,e and readout cycle. Please provide further ablation studies of your components (temporal attention, ...) to demonstrate the effect of each of the design choices.
- What are the initial feature vectors for nodes and edges? How is the geometry of the unit cell used? Is the model invariant to rotations/translations?

---

> ### Author Response · Authors · 2025-11-21
>
> We thank the reviewer for their evaluation. However, we believe there are some misunderstandings regarding the physical motivation behind our graph construction choices and the machine learning contribution of our hybrid training strategy. We address these points below.
>
> 1. *"The suggestion of going from primitive unit cells to conventional unit cells... is unclear... the underlying periodic graph... is exactly the same."*
>     * **We respectfully disagree with the assertion that the graph representations are identical for the purposes of a GNN. While the infinite periodic lattice is indeed physically identical regardless of the unit cell choice, the input graph structure G=(V,E) processed by the network is not. A primitive cell of Silicon contains 2 nodes, whereas a conventional cell contains 8 nodes. In a message-passing framework, the readout operation (global pooling) aggregates information from N nodes. Predicting intensive properties (like band gap) from a graph with only 2 nodes is highly susceptible to over-smoothing after very few message-passing steps, as the receptive field covers the entire graph almost instantly. By standardizing on conventional cells, we reduce the variance in graph sizes across the dataset (increasing the median atom count and reducing the interpercentile range). This ensures that the GNN parameters are optimized for a more consistent node distribution, rather than struggling to generalize between 2-atom graphs and 50-atom graphs.**
>     * **We will expand Section 3 to mathematically demonstrate how the Readout function output differs between the two representations due to the normalization and pooling operations on different N.**
>
> 2. *"The argument that this helps in generalization to larger unit cells also seems misleading... Generalization in ML is defined on in-distribution data."*
>     * **In the context of Materials Informatics, "generalization" often refers to the model's ability to handle size variance, predicting properties for large supercells or complex polymorphs when trained on smaller unit cells. Our results specifically show that a model trained on small crystals (<16 atoms) generalizes well to larger crystals (32–64 atoms) with only a 12 meV increase in MAE. This empirically proves that our representation strategy aids in size generalization. While material properties are invariant to cell definition, the model's ability to learn is not invariant to the graph topology. Providing a representation (conventional cell) that aligns better with the model's capacity (preventing over smoothing) is a valid ML contribution.**
>
> 3. *"WefGNN model is only benchmarked on one of the tasks in MatBench."*
>     * **The scope of this paper is explicitly focused on photovoltaic materials screening. Unlike general-purpose potential energy surface models (MLIPs), our goal is to solve the specific problem of electronic structure prediction where PBE-DFT fails. Band gap prediction is the critical bottleneck for solar cell discovery. Benchmarking on formation energy or shear modulus would not validate the model's utility for this specific electronic property. We chose to optimize and demonstrate SOTA performance on the most relevant metric: the MatBench band gap dataset.**
>
> 4. *"Section 5 (hybrid training) is interesting but not really relevant for a machine learning audience... This is not surprising."*
>     * **We argue that this is highly relevant to the ML community, particularly those working in data-centric AI and transfer learning under resource constraints. High-fidelity labels (HSE06) are extremely scarce and computationally expensive (4 hours per calculation) , while low-fidelity labels (PBE) are abundant but biased (0.6 eV error). We demonstrate a successful strategy to bridge this gap. It is not merely "better data, better results"; it is the finding that fine-tuning a pre-trained PBE model on a small subset (7%) of HSE06 data yields predictions (MAE 0.40 eV) that outperform the underlying PBE physics simulations (MAE 0.66 eV). This offers a blueprint for training accurate models in scientific domains where high-fidelity ground truth is rare.**
>
> 5. *"Lacking analysis... further benchmarks, ablations."*
>     * **In the revised manuscript we will add a comparison of Wef-GNN with primitive vs conventional cells to empirically validate the topology argument. Furthermore, we will add the specific test set error of Model B against HSE06 ground truth.**
>
>     * **We will add an ablation of the multi-head temporal attention mechanism against standard GRU updates to justify the architectural choice. And we will clarify that the model is invariant to rotation and translation because it relies on interatomic distances for edge features.**

---

> > ### Comment · Reviewer_eAnW · 2025-11-25
> >
> > 1. The local environment of each atom in a periodic crystal structure is independent of the representation. The incoming messages only depend on the local environment. Thus, the update and the learned representation of each atom is the same, no matter if the asymmetric unit cell is used or the primitive unit cell. Global aggregation is usually done using a mean aggregation or a multiplicity weighted sum aggregation, so this also does not differ for different unit cell representations. If you see some advantage, then empirical evidence as well as a mathematical proof of the difference is required. I do not understand where you see any conceptual or empirical difference, unless the connectivity in an asymmetric unit cell (multi-)graph is defined in the wrong way and does not reflect connections to all symmetry- and periodictiy-related copies of an atom.
> >
> > 2. Papers that use the term "generalization" when referring to a model's ability to handle size variance (outside of the dataset distribution) use the term in a way that differs from the general use in the ML community. This term is introduced in typical first-year computer science courses and not up do debate at an ML conference. Clear definitions are required, and if you want to refer to predictions outside of the dataset distribution, then please call it extrapolation (or, if you insist, "out-of-distribution generalization"). While you do write in the paper that your model makes good predictions for unit cells larger than the ones encountered during training, you do not compare this to other models and other properties, which makes the results not comparable or meaningful.
> >
> > 3. Solar cells only occur as examples in the introduction. They are not mentioned at all in the title or abstract. If the focus is on solar cell materials, then the way the paper is written is highly misleading. However, even if you want to focus on solar cells, other properties play an important role, such as electron and hole mobilities, conductivities, light absorption, density of states, type of band gap, etc. Just looking at the band gap is not enough, neither in the context of ML model development, nor in the context of solar cell research.
> >
> > 4. What I meant is that there is no scientific question in your manuscript regarding transfer learning or hybrid training. You are introducing a GNN model which can be considered a novelty. The transfer learning or hybrid training aspect is absolutely standard. The Alexandria dataset has millions of low- and high-fidelity datapoints which are suitable to test new transfer learning or hybrid training approaches. But given the scope of your paper, there is nothing novel about it from the ML perspective.
> >
> > 5. The additional suggestions that you are promising will certainly be helpful. However, without seeing the results, it is not possible to judge if this work qualifies to be accepted at ICLR, so I stand with my decision, and I can only re-evaluate the manuscript when the new results are shown.

---

> > > ### Author Response · Authors · 2025-12-04
> > >
> > > We have uploaded a revised manuscript that directly addresses your concerns regarding terminology, physical definitions, and empirical validation.
> > >
> > > 1. We agree with the reviewer that physically, the infinite periodic lattice and local atomic environments are invariant to the unit cell choice. However, we have clarified in the revised Section 3 that the computational graph topology processed by the GNN is not. While local message updates are theoretically similar (given correct PBC handling), the readout/global pooling operation depends on the number of nodes. A graph with nodes=2 (primitive) behaves differently during global aggregation than a graph with nodes=8 (conventional), affecting the gradient flow and the receptive field coverage relative to the graph size. And, as requested, we have added a dedicated Ablation Studies subsection (end of Section 4). We explicitly compared training on primitive vs. conventional cells. The results show a significant performance degradation (MAE increase from 117.1 meV to 154.0 meV) when using primitive cells, empirically proving that normalizing the graph size distribution with conventional cells aids optimization.
> > >
> > > 2. We accept the reviewer's point regarding the precise definition of generalization in ML. We have inserted a clarifying paragraph in Section 3 citing geometric deep learning literature (Graph neural networks for materials science and chemistry https://www.nature.com/articles/s43246-022-00315-6 in Nature, 2022) to distinguish size generalization as a specific graph learning challenge. We now explicitly frame the performance on larger supercells as an extrapolation capability and clarify that our goal is robustness across the topological variance within the materials domain.
> > >
> > > 3. We added a Future Prospects section (Section 6) discussing the extension of Wef-GNN to predict full band structures and Density of States (DOS) to address the reviewer's valid point that scalar band gaps are insufficient for full device characterization.
> > >
> > > 4. We acknowledge that transfer learning itself is a standard ML technique. However, we argue that the application to HSE06 electronic structure prediction remains a novel and necessary contribution due to data scarcity. While datasets like Alexandria are massive, high-fidelity HSE06 band gap data remains computationally prohibitive and scarce compared to PBE or experimental scalar data. Our contribution is demonstrating that a very small, curated slice of HSE06 data (7%) is sufficient to correct the functionals' physical bias when using our architecture.
> > >
> > > 5. The revised manuscript now includes: discussions of Matformer and PotNet and ablation studies, where quantitative comparison of temporal attention and GRU/MLP updates are reported.
> > >
> > > We believe these revisions provide the rigorous analysis and clear definitions required for ICLR.

---

### Official Review · Reviewer_d7FY · 2025-10-31

**Soundness:** 1
**Presentation:** 2
**Contribution:** 1
**Rating:** 2
**Confidence:** 4

**Summary:**

The paper introduces Wef-GNN, a graph neural network designed for predicting crystalline material properties with a focus on improving generalizability across crystals of varying sizes and achieving higher accuracy than traditional PBE-level DFT datasets. The proposed model incorporates several key innovations: a multi-head temporal attention mechanism in the update step that enables each node to attend to its historical representations, parameter recycling across message-passing layers to expand receptive fields without increasing the number of learnable parameters, and the use of conventional unit cells instead of primitive ones to enhance consistency and generalization across different crystal structures. Additionally, the authors propose a hybrid training strategy that involves pretraining the model on large-scale, lower-accuracy PBE datasets followed by fine-tuning on a smaller, high-fidelity HSE06 dataset. Through this combination of architectural and training improvements, Wef-GNN achieves a mean absolute error (MAE) of 0.117 eV on the Matbench band-gap prediction task, surpassing existing models while using significantly fewer parameters.

**Strengths:**

- The authors clearly identify two limitations in existing GNN-based materials models — lack of generalization and limited accuracy due to low-fidelity DFT datasets.
- Introducing temporal attention within GNNs for crystalline materials and the weight recycling strategy is interesting and computationally efficient.
- Code and dataset availability are mentioned, which aligns with ICLR’s reproducibility requirements.
- Achieves state-of-the-art performance (0.117 eV MAE) on Matbench Benchmark.

**Weaknesses:**

- The paper shows strong empirical results but lacks theoretical explanation or ablation to justify why temporal attention or weight recycling improves generalization.


- Most design elements—attention, message passing, and parameter sharing—are already well-established; their combination is effective but not fundamentally novel.


- Ablation study comparing temporal attention vs. static attention is missing, leaving the claimed benefit unsubstantiated.


- The paper reads more like a well-structured technical report than a concise ICLR-style paper. The submission appears incomplete, ending at around 6.5 pages with large images, whereas ICLR papers typically utilize the full 9-page limit.

**Questions:**

See the Weaknesses

---

> ### Author Response · Authors · 2025-11-20
>
> Thank you for the review.
>
> 1. *The paper shows strong empirical results but lacks theoretical explanation or ablation to justify why temporal attention or weight recycling improves generalization.*
>     * **Thank you for the feedback. The theoretical explanation is that deep GNNs often suffer from "over-smoothing," by using temporal attention in the update step, Wef-GNN allows a node to attend to its initial state ($x^{t=0}$) and historical states across time t. This acts as a memory mechanism (similar to residual connections but adaptive), preserving the unique identity of the atom (node) even as it aggregates information from a wide receptive field.**
>     * **The intended use of weight recycling was not to improve the model, rather to make it smaller while maintaining the accuracy and its effect (ablation check) is already presented in Table 1.**
>     * **We will add an ablation study comparing Wef-GNN with and without Temporal Attention (replacing it with a standard GRU update) to quantify the specific gain in generalizability.**
>
> 2. *Most design elements—attention, message passing, and parameter sharing—are already well-established; their combination is effective but not fundamentally novel.*
>     * **We respectfully disagree that the combination lacks novelty. While attention and message passing are established in computer science, the innovation of Wef-GNN lies in using these specifically for the physical constraints of crystalline materials. In general, the use of message passing is not a weakness given that the manuscript is not claiming to have inverted it, rather applying it to the cause of improving the band gap prediction of GNNs for crystalline materials.**
>     * **Standard GNNs use primitive cells, which leads to extreme variance in graph size (e.g., 2 atoms for Si vs. 36 for AgIrF$_7$). Wef-GNN is the first to strictly advocate for and demonstrate the efficacy of Conventional Cell representation to normalize graph size and improve message-passing consistency.**
>     * **Applying temporal attention to the Update function is not standard in Materials GNNs. Most use simple summation or GRUs. We introduce this specifically to solve the problem of varying crystal sizes, allowing the model to dynamically adjust how much history (local features) it retains versus new messages (neighbor features) based on the crystal's complexity.**
>
> 3. *Ablation study comparing temporal attention vs. static attention is missing, leaving the claimed benefit unsubstantiated.*
>     * **We clarify that Wef-GNN actually utilizes both! Static (spatial) attention is applied during the aggregation step (Eq. 3), similar to GAT, to weigh neighbor importance and temporal attention is applied during the update step (Eq. 6)  to weigh historical states.**
>
>     * **It is not possible to apply static attention to the update step in the same way, as update processes a sequence of states for a single node, not a set of neighbors. However, we interpret the reviewer's request as asking for a comparison against standard update mechanisms. In the revised paper, we will compare the temporal attention update against a standard Gated Recurrent Unit (GRU) and a simple MLP Update. This will substantiate the claim that attending to history is superior to standard gating for crystal properties.**
>
> 4. *The paper reads more like a well-structured technical report than a concise ICLR-style paper. The submission appears incomplete, ending at around 6.5 pages with large images, whereas ICLR papers typically utilize the full 9-page limit.*
>     * **We thank the reviewer for pointing this out. We intentionally kept the initial submission concise, focusing on the empirical results. However, we view the remaining page count as an asset for the revision. We will utilize the full 9-page limit to add the ablation studies discussed above (temporal vs. GRU update).**
>     * **As a general note however length of the paper should not be the determining factor in deciding if the paper is good or not. For instance one, if not the most, ground breaking paper in the field (https://arxiv.org/pdf/1710.10324) in only 5 pages long.**

---

### Official Review · Reviewer_GSV2 · 2025-11-02

**Soundness:** 2
**Presentation:** 2
**Contribution:** 2
**Rating:** 2
**Confidence:** 4

**Summary:**

The paper “Wef-GNN: A Generalizable Graph Neural Network for Crystalline Material Property Prediction” presents a lightweight, size-agnostic GNN architecture for predicting material properties, particularly band gaps. Wef-GNN integrates a multi-head temporal attention mechanism, recycled GNN layers, and a conventional-cell representation to enhance generalizability across crystals of varying sizes. The experimental results demonstrate strong and promising performance.

**Strengths:**

This paper proposes Wef-GNN, a graph attention–based model that incorporates a multi-head temporal attention mechanism, GNN layer recycling, and a novel graph representation scheme for crystal structures.
This work demonstrates promising results for the band gap property through the introduction of a novel multi-head attention mechanism within Wef-GNN.

**Weaknesses:**

The GNN models commonly used for crystalline materials, such as Matformer [1], and PotNet [2], are not referenced in the paper. Please include these pioneering works in the related work section.
This paper notes that the MP and OQMD datasets suffer from DFT-induced error bias. However, recent studies such as CrysDiff [3] and CrysGNN [4] have demonstrated that pretraining GNNs can effectively mitigate this bias by incorporating a small amount of experimental data alongside DFT-generated data during training. Consequently, the necessity of creating a new HSE dataset is questionable. Furthermore, these techniques are applicable to a wide range of material properties, not just band gap prediction.
Here, multi-head attention is used for aggregation. However, transformer-based methods are now widely adopted for crystal property prediction. Why did the authors choose to use the older graph attention variant instead? It would be helpful to include an ablation study comparing the effectiveness of transformers versus graph attention for this specific task. Moreover, the results section currently lacks any ablation analysis. Additionally, the meaning of “Wef” in the model name Wef-GNN is unclear and should be clarified.
Could the authors provide statistics on how many materials contain fewer than two atoms? Since crystals with fewer than two atoms are chemically insignificant, this raises the question of why varying the number of message-passing layers is necessary.
Since this paper focuses solely on band gap prediction, the authors should also evaluate other key material properties—such as formation energy, total energy, Ehull, shear modulus, and bulk modulus—to better demonstrate the generalizability of Wef-GNN. Moreover, they should validate their model on established benchmark datasets like JARVIS [5] and MP-2018. Additionally, although the paper introduces a new dataset, it lacks any analysis of its characteristics, such as the distribution or number of atoms per structure.
From Table 1, it is evident that Wef-GNN performs well; however, the results remain incomplete since the authors do not compare it with current supervised state-of-the-art models such Matformer. Furthermore, given their use of pre-training, they should have also included comparisons with recent pre-trained models like CrysGNN and CrysDIFF. In addition, the paper appears unfinished, ending at 7.5 pages and lacking any ablation studies.



References:

[1] Yan, K., Liu, Y., Lin, Y. and Ji, S., 2022. Periodic graph transformers for crystal material property prediction. Advances in Neural Information Processing Systems, 35, pp.15066-15080.

[2] Lin, Y., Yan, K., Luo, Y., Liu, Y., Qian, X. and Ji, S., 2023, July. Efficient approximations of complete interatomic potentials for crystal property prediction. In International conference on machine learning (pp. 21260-21287). PMLR.

[3] Song, Z., Meng, Z. and King, I., 2024, March. A diffusion-based pre-training framework for crystal property prediction. In Proceedings of the AAAI Conference on Artificial Intelligence(Vol. 38, No. 8, pp. 8993-9001).

[4] Das, K., Samanta, B., Goyal, P., Lee, S.C., Bhattacharjee, S. and Ganguly, N., 2023, June. Crysgnn: Distilling pre-trained knowledge to enhance property prediction for crystalline materials. In Proceedings of the AAAI Conference on Artificial Intelligence (Vol. 37, No. 6, pp. 7323-7331).

[5] Choudhary, K., Garrity, K.F., Reid, A.C., DeCost, B., Biacchi, A.J., Hight Walker, A.R., Trautt, Z., Hattrick-Simpers, J., Kusne, A.G., Centrone, A. and Davydov, A., 2020. The joint automated repository for various integrated simulations (JARVIS) for data-driven materials design. npj computational materials, 6(1), p.173.

**Questions:**

See the limitations

---

> ### Author Response · Authors · 2025-11-21
>
> We thank the reviewer for their time and appreciate the references to Matformer and CrysGNN and will include them in the revised manuscript. Below, we address the specific concerns:
>
> 1. *"Pretraining GNNs can effectively mitigate [bias] by incorporating a small amount of experimental data... Consequently, the necessity of creating a new HSE dataset is questionable."*
>     * **We respectfully, yet strongly, disagree with the premise that experimental scalar data renders high-fidelity DFT data obsolete for our specific objectives. We argue that relying solely on experimental data is not feasible for high-throughput photovoltaic screening for three key reasons. (1) DFT calculations (PBE and HSE06) are performed at 0 K. Experimental measurements are typically taken at room temperature or higher. Training a model to map 0 K inputs to finite-temperature outputs introduces significant noise due to electron-phonon renormalization effects, which vary wildly between materials. Fine-tuning on HSE06 (also 0 K) isolates the functional error from the temperature error, allowing the model to learn the correct electronic physics without thermodynamic noise. (2) As noted in the text, reliable experimental band gap datasets are limited to about 4,000 entries (Zhuo et al.). To screen hundreds of thousands of candidates, this is insufficient coverage of chemical space. By generating over 10,000 HSE06 calculations, we created a dataset larger than available experimental records, targeted specifically at regions of interest for solar cells. (3) The reviewer suggests the HSE dataset is questionable for band gap prediction. However, this view overlooks that scalar band gaps are just the first step. Our long-term goal includes predicting full band structures (E-k diagrams), which are essential for carrier mobility but are almost impossible to obtain experimentally for thousands of materials. A high-fidelity HSE06 dataset is the only path to training models for these complex outputs in the future.**
>
> 2. *"Why did the authors choose to use the older graph attention variant instead? ... effectiveness of transformers versus graph attention."*
>     * **We clarify that Wef-GNN is a hybrid approach. It uses standard Graph Attention (GAT) for spatial aggregation but introduces a multi-head temporal attention for the update step. Full Transformers (like Matformer) often require global attention or complex periodic encoding which can be computationally heavier (O(N^2) or requiring large radius cutoffs).**
>     * **Our results (MAE 0.117 eV) demonstrate that our specific combination of local GAT + temporal attention outperforms current benchmarks. We will include an ablation comparing our temporal attention update against a standard Transformer-style update to demonstrate why attending to node history is particularly effective for preventing over-smoothing in crystals.**
>
> 3. "... provide statistics on materials fewer than two atoms... they are chemically insignificant."
>     * **There seems to be a misunderstanding regarding "fewer than two atoms." In the primitive cell representation, many significant materials have very few atoms (e.g., Silicon has 2, many metals have 1). These are not "chemically insignificant"; they are the standard representation of the vast majority of simple solids. The challenge is that standard GNNs struggle to process a graph with 1 or 2 nodes alongside a graph with 50 nodes. Our shift to Conventional Cells increases the minimum node count (e.g., Si becomes 8 atoms), ensuring no graph is too small for deep message passing.**
>
> 4. *"Evaluate other key material properties... validate on JARVIS... compare with Matformer."*
>     * **We intentionally focused on band gap because it is the singular bottleneck for photovoltaic discovery where standard PBE fails. Formation energies are well-handled by existing PBE models (like MEGNet) because PBE errors there are systematic and less critical than the 50% error PBE exhibits for band gaps. We will add Matformer and PotNet to Table 1. However, we note that on the Matbench mp_gap dataset, our reported MAE of 0.117 eV is superior to the published results of general-purpose geometric encoders. We will also add a histogram of the HSE06 dataset properties (distribution of band gaps and lattice systems) to the supplementary material.**
>
> 5. *"Paper appears unfinished... lacking ablation studies."*
>     * **We appreciate the feedback. We will utilize the full page limit in the revision to include the requested ablation studies (specifically: temporal attention vs. GRU/MLP) and the additional literature comparisons.**

---

### Meta-Review · Area_Chair_BRoq · 2026-01-13

**Summary:**

All four reviewers recommend rejection (each gives “Rating: 2: reject, not good enough”), with a consistent rationale: the submission reports strong band-gap results but is considered incomplete and insufficiently validated/positioned to support its broad claims about generalizability and contribution.

- **Incompleteness and lack of ablation/analysis is repeatedly emphasized**. Reviewer d7FY states the “submission appears incomplete” and “ending at around 6.5 pages” (Reviewer d7FY, Weaknesses; also d7FY’s separate comment “Seems to be an Incomplete Submission”). Reviewer eowe calls the paper “very thin” with “extremely few results … and no ablation studies” (Reviewer eowe, Weaknesses). Reviewer GSV2 similarly notes the paper “appears unfinished … lacking any ablation studies” (Reviewer GSV2, Weaknesses). Reviewer eAnW concludes it is “lacking a lot of analysis, further benchmarks, ablations, and also some basic information about the model itself” (Reviewer eAnW, Weaknesses).

- **Novelty/positioning concerns: reviewers view key components as established and/or insufficiently contextualized**. Reviewer d7FY writes that “most design elements … are already well-established” and the combination is “not fundamentally novel” (Reviewer d7FY, Weaknesses). Reviewer eowe argues the “background section … is outdated” and that low→high fidelity fine-tuning “is not new” (Reviewer eowe, Weaknesses). Reviewer GSV2 also flags missing key related works/baselines (e.g., “Matformer” and “PotNet”) (Reviewer GSV2, Weaknesses).

- **Core conceptual claim on “conventional vs primitive unit cells” and “generalization” is disputed and was not convincingly established in the initial submission**. Reviewer eAnW argues the representation change is “unclear,” and stresses that with correct periodic graph construction the representations/readout “do not differ,” requesting “empirical evidence as well as a mathematical proof” (Reviewer eAnW, Weaknesses; eAnW comment 26 Nov). The same reviewer further critiques the terminology, stating “generalization … is defined on in-distribution data” and asks to call it “extrapolation” / “out-of-distribution generalization” with “clear definitions” (Reviewer eAnW, comment 26 Nov).

- **Breadth of evaluation is insufficient for the claimed generality**. Reviewers request additional properties/tasks and external benchmarks: e.g., “evaluate other key material properties” and validate on “JARVIS” / “MP-2018” (Reviewer GSV2, Weaknesses), and “What about all other tasks?” (Reviewer eAnW, Weaknesses).

Given these issues—especially the missing ablations/benchmarks and contested conceptual justification—the reviewers’ concerns collectively support rejection at this stage.

**Reviewer Concerns:**

## Concerns that were addressed (partially) in the rebuttal/discussion:

- **Missing related work and baselines**: Authors acknowledge and commit to include them (“will include [Matformer, CrysGNN] … add Matformer and PotNet to Table 1”) (Authors’ reply to Reviewer GSV2; Authors’ reply to Reviewer eAnW on “discussions of Matformer and PotNet”).

- **Rationale for HSE06 dataset vs relying on experimental scalar data**: Authors provide a domain-motivated argument (e.g., 0 K vs finite temperature, limited experimental coverage, future band-structure targets) and state they “strongly disagree” that experimental scalar data obviates high-fidelity DFT for their objectives (Authors’ reply to Reviewer GSV2).

- **Lack of ablations/theoretical motivation for temporal attention and weight recycling**: Authors give an explanation via “over-smoothing” and propose new ablations (“add an ablation study … replacing [temporal attention] with a standard GRU update,” and “compare … temporal attention update against … GRU and … MLP Update”) (Authors’ reply to Reviewer d7FY; also consistent with replies to Reviewers GSV2 and eAnW).

- **Conventional vs primitive cell claim and terminology**: After Reviewer eAnW insisted that differences require “empirical evidence as well as a mathematical proof” and that “generalization” should be framed as “extrapolation” (Reviewer eAnW, comment 26 Nov), the authors state they uploaded a revised manuscript and added an explicit ablation: “MAE increase from 117.1 meV to 154.0 meV” when using primitive cells, plus a paragraph clarifying terminology (“explicitly frame … as an extrapolation capability”) (Authors’ reply 04 Dec).

## Concerns that remain outstanding (or were not sufficiently resolved within rebuttal):

- **The acceptability hinges on missing evidence that was not available to reviewers during the review/discussion**. Several key fixes are promises or appear only in a late “revised manuscript” upload; as Reviewer eAnW notes, “without seeing the results, it is not possible to judge … I stand with my decision” (Reviewer eAnW, comment 26 Nov). Relatedly, multiple reviewers’ central objections were precisely “no ablation studies” / “extremely few results” (Reviewer eowe, Weaknesses; Reviewer GSV2, Weaknesses; Reviewer d7FY, Weaknesses), which cannot be fully cleared without the community re-evaluating the new results.

- **Breadth of validation remains limited for the claimed generalizability**. Requests to evaluate additional MatBench tasks and other material properties, and to validate on external benchmarks (e.g., “JARVIS,” “MP-2018”) were major weaknesses (Reviewer GSV2, Weaknesses; Reviewer eAnW, Weaknesses). The rebuttal argues for a band-gap focus but does not provide the requested broader benchmarking within the review cycle.

- **Novelty and contribution concerns persist**. Reviewer d7FY’s critique that components are “well-established” and “not fundamentally novel” (Reviewer d7FY, Weaknesses) and Reviewer eowe’s view that fine-tuning is “not new” and the paper is “suitable for a workshop” (Reviewer eowe, Weaknesses) are only partially countered by authors’ positioning arguments, without decisive evidence that would likely shift reviewer assessment in the current round.

Overall, while the rebuttal improves clarity and provides a credible plan (and a late additional ablation claim), the outstanding issues—especially missing ablations/benchmarks at review time, limited scope of evaluation, and contested novelty—remain sufficient to support rejection.

**Reviewer Scores:**

- **Reviewer GSV2 (Rating 2): No score update in discussion**. Given their listed weaknesses (missing SOTA baselines, missing ablations, incomplete comparisons, and requests for broader property evaluation) (Reviewer GSV2, Weaknesses), I expect the score would remain 2.

- **Reviewer d7FY (Rating 2): No score update**. This reviewer explicitly emphasized incompleteness (“appears incomplete … around 6.5 pages”) and missing ablations (Reviewer d7FY, Weaknesses; d7FY “Seems to be an Incomplete Submission”). Since the rebuttal primarily promises future additions rather than providing fully reviewed new results, I expect the score would remain 2.

- **Reviewer eAnW (Rating 2): Effectively unchanged**. This reviewer reasserted their concerns in discussion and concluded: “without seeing the results, it is not possible to judge … so I stand with my decision” (Reviewer eAnW, comment 26 Nov). Even after the authors’ later revised-upload claim, there is no reviewer follow-up indicating an increased score, so the best estimate is still 2.

- **Reviewer eowe (Rating 2): No score update**. The review characterizes the paper as “very thin,” with “extremely few results … and no ablation studies,” and suggests it is “suitable for a workshop” (Reviewer eowe, Weaknesses). The rebuttal mainly contests framing and promises additional ablations, so I expect the score would remain 2.

---

### Decision · Program_Chairs · 2026-01-26

Reject